# A Dual-Mode Method Based on Aptamer Recognition and Time-Resolved Fluorescence Resonance Energy Transfer for Histamine Detection in Fish

**DOI:** 10.3390/molecules27248711

**Published:** 2022-12-09

**Authors:** Xin Wang, Fu Yang, Chengfang Deng, Yujie Zhang, Xiao Yang, Xianggui Chen, Yukun Huang, Hua Ye, Jianjun Zhong, Zhouping Wang

**Affiliations:** 1School of Food and Biological Engineering, Xihua University, Chengdu 610039, China; 2Chongqing Key Laboratory of Speciality Food Co-Built by Sichuan and Chongqing, Chengdu 610039, China; 3School of Grain Science and Technology, Jiangsu University of Science and Technology, Zhenjiang 212100, China; 4School of Food Science and Technology, Jiangnan University, Wuxi 214122, China

**Keywords:** histamine, time-resolved fluorescence, aptamer, colormetric, dual-mode detection

## Abstract

Histamine produced via the secretion of histidine decarboxylase by the bacteria in fish muscles is a toxic biogenic amine and of significant concern in food hygiene, since a high intake can cause poisoning in humans. This study proposed a fluorometric and colorimetric dual-mode specific method for the detection of histamine in fish, based on the fluorescence labeling of a histamine specific aptamer via the quenching and optical properties of gold nanoparticles (AuNPs). Due to the fluorescence resonance energy transfer phenomenon caused by the proximity of AuNPs and NaYF4:Ce/Tb, resulting in the quenching of the fluorescence signal in the detection system, the presence of histamine will compete with AuNPs to capture the aptamer and release it from the AuNP surface, inducing fluorescence recovery. Meanwhile, the combined detection of the two modes showed good linearity with histamine concentration, the linear detection range of the dual-mode synthesis was 0.2–1.0 μmol/L, with a detection limit of 4.57 nmol/L. Thus, this method has good selectivity and was successfully applied to the detection of histamine in fish foodstuffs with the recoveries of 83.39~102.027% and 82.19~105.94% for *Trichiurus haumela* and *Thamnaconus septentrionalis*, respectively. In addition, this method was shown to be simple, rapid, and easy to conduct. Through the mutual verification and combined use of the two modes, a highly sensitive, rapid, and accurate dual-mode detection method for the analysis of histamine content in food was established, thereby providing a reference for the monitoring of food freshness.

## 1. Introduction

As a heterocyclic biogenic amine produced by histidine under the action of decarboxylase, histamine is commonly found in dark meat and fish products containing Morganella morganii and Photobacterium phosphoreum [1,2,3]. Since it is highly toxic, histamine content is often used as an important index to evaluate the degree of food spoilage. Studies have proven that histamine is a potential precursor of carcinogenic N-nitroso compounds, and the excessive intake of food containing histamine can cause symptoms of poisoning such as conjunctivitis, dizziness, nausea, chest tightness, and in cases of severe reaction, even death [1,4]. Consequently, numerous countries have prescribed limits to the permissible content of histamine, depending on the specific characteristics of the food type. The United States Food and Drug Administration (FDA), for example, stipulates that the histamine content in fish should not exceed 50 mg/kg, while the European Food Safety Authority (EFSA) prescribes a histamine content limit of 100–200 mg/kg in fresh fish products. China’s food safety supervision and management department has set the limit of histamine including high histamine fish, which is not permitted to exceed 400 mg/kg, while in other sea fish, it may not exceed 200 mg/kg [5]. With the development of the foreign trade of aquatic products in China, along with the focus on improving the quality of life, food safety has gained increased attention including an awareness that the development of a sensitive and rapid technology to detect food histamine is crucial to ensure food safety.

Current histamine detection methods traditionally involve high performance liquid chromatography [6,7], capillary electrophoresis technology [8,9], enzyme-linked immunosorbent assay technology [1], and electrochemical sensing technology [10,11], among others. Although these methods do have the advantages of high sensitivity and accuracy, they also have some disadvantages such as expensive chromatographic equipment, long detection cycles, and relatively complex operation, which are not suitable for rapid detection in the field; electrophoretic techniques and enzyme-linked immunosorbent assays (ELISA) have limitations in detection volume and operational techniques and time. The portability of rapidly evolving electrochemical instruments has been applied to rapid detection in the field, but their limited temperature range and lifetime limit the popularity of electrochemical sensing techniques. The more recently developed aptamer, a type of biological molecular probe produced through a process of artificial ligand screening called the systematic evolution of ligands by exponential enrichment (SELEX) from a single-stranded oligonucleotide with a high affinity and specificity to target molecules, has been shown to have excellent potential as an alternative to antibodies [12,13]. Aptamers are currently widely used in the field of food safety detection. Sharma et al. [14] developed an aptamer assay for aflatoxin B1 (AFB1) and ochratoxin A (OTA). The developed assay platform showed high specificity and was practically applied in the detection of milk samples. Huang et al. [15] selected and characterized an aptamer against Streptococcus pyogenes and developed an aptamer-based fluorescent detection method for Streptococcus pyogenes in cooked chickens. AuNPs have attracted extensive attention in the field of chemical and biosensing because of their surface plasmon resonance, which can quickly respond to and mediate colorimetric signals [5]. Lerga et al. [16,17] used SELEX (Systematic Evolution of Ligands by Exponential Enrichment) technology to screen and obtain the histamine aptamer H_2_ and constructed a colorimetric detection method based on AuNPs in order to realize the detection of histamine in saliva, urine, blood, and food extracts.

Time-resolved fluorescence is a common detection method that uses lanthanide labeling to identify target molecules, monitors the deactivation process of excited molecules in the form of light radiation, and uses time-resolved technology to resolve signals, thereby realizing the microscopic dynamic analysis of molecules [18]. This technology avoids the background interference caused by sample autofluorescence, has the advantages of high sensitivity and easy operation, and plays an important role in food processing, food detection, and food composition analysis, among other fields [19,20,21]. In recent years, the visual and rapid detection by AuNPs based on their aggregation effect has been increasingly reported, and has been applied in fields ranging from food safety and life science analysis to cancer diagnosis and treatment. In our previous work, the time-resolved fluorescent nanomaterial NaYF_4_:Ce/Tb was synthesized using the one-step solvothermal method, followed by the successful construction of a time-resolved fluorescence-sensitive nanoprobe via the addition of aptamers to its surface [22]. However, fluorescence/colorimetric dual-mode detection technology based on AuNPs has thus far not been reported and, hence, this study aimed to establish a time-resolved fluorescence/colorimetric dual-mode detection method for histamine based on the multiple optical effects of AuNPs. This approach not only utilizes the high sensitivity of fluorescence analysis, but also the colorimetric detection advantages of simple operation and rapid reaction. Moreover, compared with the single detection mode, dual-mode detection has been shown to have higher sensitivity, a wider range of detection, and a faster detection speed, thereby providing a new mode of technical support for the rapid and accurate detection of histamine content in aquatic products.

## 2. Results

### 2.1. Feasibility of Fluorometric and Colorimetric Dual-Mode Detection of Histamine

The fluorescence results and UV–Vis absorption spectra of the histamine were used to verify the feasibility of the novel sensing strategy [23,24]. As shown in Figure 1a, the Aptamer@NaYF_4_:Ce/Tb exhibited a high initial fluorescence value. In the blank sample without histamine, the probe was adsorbed on the surface of the AuNPs. Due to the fluorescence resonance energy transfer, the fluorescence of the Aptamer@NaYF_4_:Ce/Tb was quenched, thereby greatly reducing the fluorescence signal of the reaction system. However, in the presence of histamine, the Aptamer@NaYF_4_:Ce/Tb bonded to the histamine, undergoing conformational change, and then fell away from the surface of the AuNPs, whereafter the fluorescence intensity was significantly increased.

In Figure 1b, the characteristic surface plasmon resonance (SPR) peak band of the pure AuNP solution could be observed at approximately 540 nm. Compared with the pure AuNP solution, the AuNP solution adsorbed with the aptamer showed basically no change in the position of the peak band, absorbance and color (which remained red), as shown in Figure 1c. However, after the addition of histamine, the color of the AuNP solution changed from red to blue-gray, and a strong absorption peak appeared at 640 nm, indicating the serious aggregation of the AuNPs (Figure 1b). Furthermore, the surface of the AuNPs adsorbed with the aptamer exhibited a red shift at 540 nm, and the absorption peak at 640 nm was slightly increased, implying that the aptamer could protect the AuNPs from aggregation while, at the same time, the histamine could replace citrate ions on the surface of the AuNPs through the imidazole ring, leading to partial aggregation [25,26]. In conclusion, these two effects are clear indications that the dual-mode detection method based on the AuNP aptamer is feasible.

### 2.2. Optimization of Dual-Mode Sensing System

#### 2.2.1. Optimization of Aptamer Concentration

In this experiment, different concentrations of inducer were added to 180 μL of AuNPs and incubated for 30 min at 25 °C, and then 2.0 μmol/L of histamine was added to observe the change in colorimetric signal. The results are shown in Figure 1d, and it can be seen that the absorption value of the AuNP solution and the corresponding color change of the solution were not obvious with the increase in the inducer concentration, indicating that the aggregation effect of histamine on the AuNPs was relatively weak. When the concentration of the inducer was greater than 40 nmol/L, histamine had no aggregation effect on AuNPs and the color was no longer gray, confirming the protective effect of the inducer on AuNPs, and partial histamine was needed to induce the aggregation of AuNPs for colorimetric detection. Meanwhile, the effect of different concentrations of Aptamer@NaYF4:Ce/Tb on the fluorescence recovery of AuNPs was measured using a microplate reader. As can be seen in Figure 2a, the fluorescence recovery value (*ΔF*) gradually decreased as the concentration of Ap-tamer@NaYF4:Ce/Tb increased with the same amount of AuNPs and histamine involved in changing the fluorescence intensity of the substance. When the inducer concentration was 40 nmol/L, *ΔF* reached a maximum value, and then decreased and stabilized. Therefore, based on the combined results of these two methods, the inducer concentration of 40 nmol/L was selected as the optimal dose concentration.

#### 2.2.2. Optimization of Reaction Time

The effects of different incubation times on the fluorescence recovery of the detection system after the addition of histamine were investigated. The results are presented in Figure 2b, in which it can be seen that with the addition of histamine, the fluorescence recovery value gradually increased as the reaction time extended. When the period of reaction reached 40 min, the fluorescence recovery value reached its maximum and subsequently stabilized, indicating that a stable fluorescence signal could be obtained after 40 min incubation with histamine.

### 2.3. Establishment of Fluorometric/Colorimetric Dual-Mode Method for Detection of Histamine

Based on the preliminary optimization experiments, two detection methods, the fluorescence method and colorimetric method, were used to detect histamine, and the relationship between fluorescence recovery and absorbance ratio and histamine concentration were obtained, respectively. The detection results were synthesized according to Equation (1) [27], as shown in Figure 3, with a linear range of 0.2–1.0 μmol/L, a correlation coefficient of R = 0.9996, a linear regression equation of Y = 841376X−1592, and a detection limit of 4.57 nmol/L.
(1)$$K=\frac{\Delta F}{A_{640}/A_{540}}$$

### 2.4. Precision Analysis of the Measurement Results

Precision is a general term indicating the precision of the measurement methods and results. Herein, the precision of the measurement method is denoted by the relative standard deviation of the sample (the ratio of sample standard deviation S to the sample mean $\bar{x}$) with the symbol *RSD* (relative standard deviation of the sample) = $S/\bar{x}$. According to the experimental method described in Section 3.4.5, the results are shown in Table 1 and Table 2.

The results showed that the precision range of the fluorometric recovery value was 1.4–3.8%, while the precision of the UV absorbance value was 0.1–2.8%, indicating that the precision, stability, and repeatability of the method were good.

### 2.5. Analysis of Specificity

Muscle tissue usually contains a large amount of free histidine, which, under certain conditions, is converted into histamine under the action of decarboxylase. In order to further verify the specificity and selectivity of this method, dimethylamine, trimethylamine, β-phenylethylamine, ethylenediamine, spermidine, tryptamine, and other histamine analogs (all with concentrations of 10 μmol/L) were selected for the specificity test. As can be seen from the experimental results presented in Figure 4, dimethylamine, trimethylamine, β-phenylethylamine, ethylenediamine, spermidine, and tryptamine produced no notable signal value, while the relative fluorescence intensities of histamine and histidine, dead amine and spermine were 1.75%, 7.31%, and 56.22% of the histamine, respectively. These results not only indicate that the method has good selectivity, but also that the presence and interference of spermine in the detection system should be avoided.

### 2.6. Comparison with Other Methods

Table 3 lists the detection effects of different histamine detection methods, wherein it can be seen that the detection limit of the dual-mode method constructed by this method was 4.57 nmol/L, with a linear range of 0.2–1.0 μmol/L. Furthermore, compared with both the single fluorescence method and the colorimetric method, it showed a higher degree of sensitivity. Thus, in comparison with other methods, the proposed method was found to have a wider linear range and higher detection sensitivity.

### 2.7. Sample Detection Analysis and Recovery Experiment

According to the national standard method (high performance liquid chromatography, HPLC) and the histamine detection experiment described by Huang et al. [32], the standard curve of correlation between the peak area and histamine concentration was Y = 30.09031X + 20.09641, based on a series of standard samples (histamine standard with gradient concentrations of 1–50 mg/L), while the detection limit of the national standard method was 0.0370 mg/kg. Based on the same series of samples, the concentrations of histamine in *Trichiurus haumela* and *Thamnaconus septentrionalis* determined by the national standard method were 0.0696 mg/kg and 0.0466 mg/kg, respectively, while the concentrations of histamine in the *Trichiurus haumela* and *Thamnaconus septentrionalis* determined by the repeatability condition were 0.07395 mg/kg and 0.0485 mg/kg, respectively. The absolute difference between the two independent measurements was not more than 10% of the arithmetic mean value, indicating that the national standard method had good precision. The concentrations of histamine detected by aptamer identification time-resolved fluorescence resonance energy transfer were 0.0720 mg/kg and 0.0508 mg/kg, respectively. For the same sample, the absolute difference between the two independent determination results under the condition of repeatability did not exceed 10% of the arithmetic mean value, and the two methods had a good correlation.

In order to evaluate the practical application of this method, real samples of *Trichiurus haumela* and *Thamnaconus septentrionalis* were collected for the experiment. The detection results are shown in Table 4. At the concentration of 0–10 mol/L, the recovery rates of the *Trichiurus haumela* and *Thamnaconus septentrionalis* samples were 83.41–101.94% and 82.24–105.92%, respectively. These results indicate that the dual-mode detection system developed in this study has a good recovery rate and can be used to detect histamine in *Trichiurus haumela* and *Thamnaconus septentrionalis*.

Although high performance liquid chromatography has high reproducibility, good sensitivity, and column efficiency, it requires pretreatment such as the derivatization of histamine, which is not conducive to rapid detection in situ. Based on aptamer identification time-resolved fluorescence resonance energy transfer, this experiment determined histamine simply and quickly. Moreover, this experiment verified that the performance indices met the requirements, the comparison results with the reference method GB 5009.208–2016 were satisfactory, and the established method was feasible and can be used for the determination of the histamine residue in the actual samples.

## 3. Materials and Methods

### 3.1. Materials and Reagents

Avidin, ampicillin sodium salt, Y(NO_3_)_3_ •6H_2_O, Ce(NO_3_)_3_ •6H_2_O, Tb(NO_3_)_3_ •5H_2_O were purchased from Sigma Aldrich (Shanghai, China) Trading Co., Ltd.; O-phosphorylethanolamine (AEP) was purchased from Tishi (Shanghai, China) Chemical Industrial Development Co., Ltd.; HAuCl_4_•4H_2_O (purity ≥ 99.9%) was purchased from Shanghai Aladdin Biochemical Technology Co., Ltd.; ethylene glycol solution containing appropriate amounts of NH_4_F, ethylene glycol, pentanediol (25% in *v*/*v*), HCl, NaCl, trisodium citrate (Na_3_C_6_H_5_O_7_•2H_2_O), Sinopharm Chemical Reagent Co. histamine (HTA), dimethylamine (DMA), trimethylamine (TMA), β-phenylethylamine (β-PA), ethylenediamine (ELA), spermidine (SMD), and tryptamine (TPA) were purchased from Shanghai Maclean Biochemical Technology Co., Ltd.; ultrapure water prepared by a Milli Direct-Q water purification system (Millipore Corp., Bedford, MA, USA) was used in all experiments (18.2 MΩ•cm); *Trichiurus haumela* and *Thamnaconus septentrionalis* (local supermarket).

The biotin-modified histamine aptamer sequence (5′-AGCTCCAGAAGATAAATTACAGGGAACGTGTTGGTGCGGTTCTTCCGATCTGCTGTTCTCTCTATCTGTGCCATGCAACTAGGATACTATGACCCCGG-3′) was synthesized by Bioengineering Ltd. (Shanghai, China) and purified by high performance liquid chromatography.

### 3.2. Instruments and Equipment

The following instruments and equipment were used in the analysis: A laser particle size meter (ZSN3600), Malvern Instruments Ltd., Malvern, UK; fluorescence spectrometer (FLUOROMAX-4CP), HORIBA Corporation, Irvine, CA, USA; double beam ultraviolet spectrophotometer (SDPTOPUV2800PC), Shanghai Sunyu Hengping Scientific Instruments Co., Ltd. (Shanghai, China); small high-speed centrifuge (5424), Eppendorf AG, Hamburg, Germany; vacuum drying oven (BZF-50), Shanghai Boxun Industrial Co., Ltd., Shanghai, China; pure/ultrapure water all-in-one machine (Direct-Q), Millipore Corporation, Burlington, MA, USA; and JEM-2100 high-resolution transmission electron microscope (Nippon Electron Co., Ltd. Tokyo, Japan).

### 3.3. Detection Principle

The principle of histamine detection applied in this study is shown in Figure 5. First, the histamine aptamer was immobilized on NaYF_4_:Ce/Tb nanofluorescent material through biotin–avidin interaction. The subsequent addition of unmodified AuNPs to the system triggered the fluorescence resonance energy transfer phenomenon, resulting in the quenching of the fluorescence signal. When the target histamine was added, it competed with the AuNPs to bind to the aptamer because of the high affinity of its aptamer to the target histamine [33]. The AuNPs were, thus, separated from the aptamer and released into the solution, enabling the recovery of the fluorescence signal. Finally, the dual-mode detection of histamine was realized via the measurement of the degree of time-resolved fluorescence recovery and linear analysis of the histamine concentration. The colorimetric method replaces the citrate ion on the surface of synthetic AuNPs by a special imidazole ring on the surface of the histamine. The combination of positively charged histamine and negatively charged citrate ions reduces the net surface charge of AuNPs, which destabilizes AuNPs and triggers aggregation, and the color of the solution changes from red to blue-gray, so the presence of histamine can also be detected by visual observation. Moreover, AuNPs have a characteristic absorption peak at 540 nm, and when aggregation occurs, the intensity of the absorption peak at 540 nm decreases and a new absorption peak appears at 640 nm, which is due to the red-shift of the surface plasmon resonance peak of the aggregates. Therefore, the concentration of the target histamine was analyzed by measuring the absorbance at 540 and 640 nm and determining the degree of aggregation of AuNPs by its ratio (A640/A540). Accurate quantification and simple, rapid determination of the histamine content was conducted by dual-mode fluorescence and colorimetric methods.

### 3.4. Experimental Methods

#### 3.4.1. Preparation of AuNPs

Add l mL HAuCl_4_•4H_2_O solution (1 wt%) and 99 mL ultrapure water to a round bottom flask, heat to boiling under magnetic stirring, then add 2 mL 1% aqueous sodium citrate solution (weigh 0.5 g sodium citrate solid into 100 mL beaker and add 50 mL deionized water, stir to dissolve); the reaction is kept boiling with continuous stirring for 10 min, after which it is turned off. The heating power was turned off to stop the heating, and then the stirring was continued for 15 min, and the solution was cooled to room temperature to finally obtain a transparent wine red solution, which was placed in a brown reagent bottle and stored at 4 °C for use (the prepared AuNPs were burgundy in color and characterized with transmission electron microscopy) (Appendix A).

#### 3.4.2. Preparation of NaYF4:Ce/Tb Nanoparticles

The nanomaterials of NaYF_4_:Ce/Tb were synthesized by referring to the method of Wang et al. [34]. In a round bottom flask, 1.0 mmol AEP and a certain stoichiometric ratio of NaCl, Y(NO_3_)_3_•6H_2_O, Ce(NO_3_)_3_•6H_2_O, and Tb(NO_3_)_3_•5H_2_O were added and dissolved with ethylene glycol solution containing an appropriate amount of NH_4_F and aged for 30 min, transferred to an autoclave at 180 °C for 4 h, and removed and cooled naturally to room temperature. After washing and drying in a vacuum drying oven at 60 °C for 10 h, the solid particles were stored at room temperature and protected from light (scanning time-resolved fluorescence of the synthesized NaYF4:Ce/Tb nanofluorescent materials and characterization by transmission electron microscopy).

#### 3.4.3. Preparation of NaYF4:Ce/Tb Nanoparticles Modified with Avidin

The classical glutaraldehyde method was used to couple the avidin with the amino groups on the surface of NaYF_4_:Ce/Tb. A total of 5 mg of NaYF_4_:Ce/Tb nanoparticles were weighed and dispersed in 1 mL of phosphate buffered saline (PBS), sonicated for 10 min, the solution was reacted with 5 mL of glutaraldehyde aqueous solution, and shaken slowly at room temperature for 2 h. The remaining glutaraldehyde reaction solution was discarded and washed three times with phosphate buffer, and a certain amount of avidin was added overnight at 37 °C. The reaction was followed by centrifugation and washing with PBS three times. Finally, the samples were resuspended in 5 mL PBS and stored at 4 °C for later use.

#### 3.4.4. Assembly of Aptamer@NaYF4:Ce/Tb Nanoparticles

The quantified aptamers were added into the avidin-based nanoparticle suspension and incubated at 37 °C for 3 h with slow shaking. After the reaction, it was centrifuged and washed three times with PBS, resuspended in PBS, and stored at 4 °C.

#### 3.4.5. Construction of a Fluorescent Colorimetric Dual-Mode Based Method for the Detection of Histamine

In the detection system, different concentrations of histamine solutions were detected with a certain concentration of fluorescent probe. The target histamine was incubated with the fluorescent detection probe at 25 °C with shaking for 0.5 h. The fluorescence intensity of the solution was measured after the reaction. The measurement conditions were set as follows: excitation wavelength 273 nm, delay time 0.1 ms, detection time 1 ms.

The synthesized AuNPs (180 μL) were mixed with the histamine aptamer (30 μL, 40 nmol/L) so that the aptamer was uniformly adsorbed on the surface of the AuNPs. Different concentrations of histamine solutions were added, mixed well, and incubated at room temperature (25 °C) for 30 min before UV–Vis spectroscopy scanning to evaluate the degree of AuNP aggregation by the ratio of the absorbance value at 640 nm to that at 540 nm. Meanwhile, the fluorescence intensity of the supernatant was measured using an equal volume of PBS buffer instead of histamine as a blank control with an excitation wavelength of 273 nm and emission wavelength of 545 nm, and the fluorescence difference (*ΔF*) between the experimental and blank groups was calculated. Finally, the measurement values of the two methods were combined to produce the final data K, which is the ratio of the fluorescence recovery value to the signal change in absorbance at wavelengths of 640 nm and 540 nm [27].

#### 3.4.6. Spiked Recoveries of Actual Samples with Detection Applications

Referring to the method of Lerga et al. [16], the flesh of *Trichiurus haumela* and *Thamnaconus septentrionalis* was picked, and the fish was minced for subsequent experiments. A total of 0.5 g of fish mince was weighed, 25 mL ultrapure water was added, and the supernatant was collected after 1~2 min of vortex oscillation, and then centrifuged at 10,000 r/min for 5 min. Histamine standard solutions of different concentrations were added to the fish extract supernatant, and three standard addition and recovery experiments were performed for each solution according to the method in Section 3.4.4, and the recoveries were calculated.

## 4. Conclusions

A simple and rapid dual-mode method for the detection of histamine was proposed by using a histamine aptamer modified with fluorophore (aptamer@NaYF4:Ce/Tb), which was conducted in fluorometric and colorimetric dual-mode method through taking the AuNPs as carriers and the time-resolved fluorescence nanoparticle modified aptamer as the recognition element. The results showed that the changes in fluorescence intensity and absorbance ratio were linearly related to the histamine concentration, showed good selectivity for histamine, and was successfully applied to histamine in fish meat. In addition, the method does not require the action of enzymes, the instrumentation and operating procedures used are relatively simple and largely independent of the environment, and the unique biocompatibility and optical properties of AuNPs can be utilized for their powerful signal generation and transduction capabilities, which contribute to the rapid and efficient reactions. By mutual validation comparison, this dual-mode achieved a satisfactory analytical performance for complex food matrix samples, avoiding the single detection. The dual mode has great potential for development and application in the field of food decontamination by avoiding possible interference in a single detection mode. The implementation of multiple detection methods with joint application and mutual validation of the diversified detection mode is important to achieve real-time, simultaneous monitoring of hazardous substances in complex samples in the field. Currently, this dual-mode detection method can only detect a single histamine, and has limitations in detecting multiple biogenic amines with the widely varying types and levels in food.

## Figures and Tables

**Figure 1 molecules-27-08711-f001:**
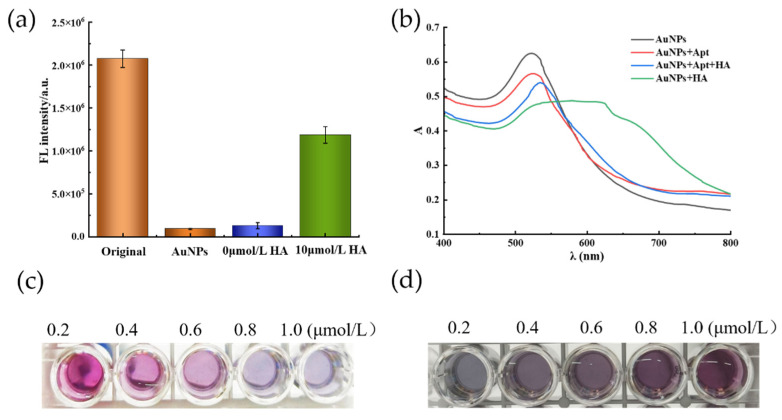
(**a**) Fluorescence signal of the histamine aptamer in different systems. (**b**) The UV–Visible absorption spectrogram of detection principle feasibility (histamine concentration of 10 μmol/L). (**c**) Colorimetric results of different histamine concentrations in the same volume of AuNP solution at the same aptamer concentration (from left to right histamine concentrations in the order of 0.2–1.0 μmol/L). (**d**) Colorimetric results of histamine at 2.0 µmol/L concentration in solutions with different aptamer concentrations and AuNPs (from left to right aptamer concentration of 20–100 nmol/L).

**Figure 2 molecules-27-08711-f002:**
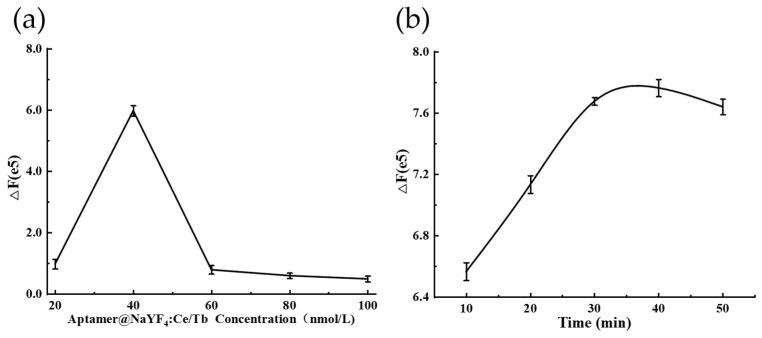
(**a**) Fluorescence recovery values of histamine at different Aptamer@NaYF_4_:Ce/Tb concentrations. (**b**) Fluorescence recovery value of Aptamer@NaYF_4_:Ce/Tb after different periods of reaction with histamine.

**Figure 3 molecules-27-08711-f003:**
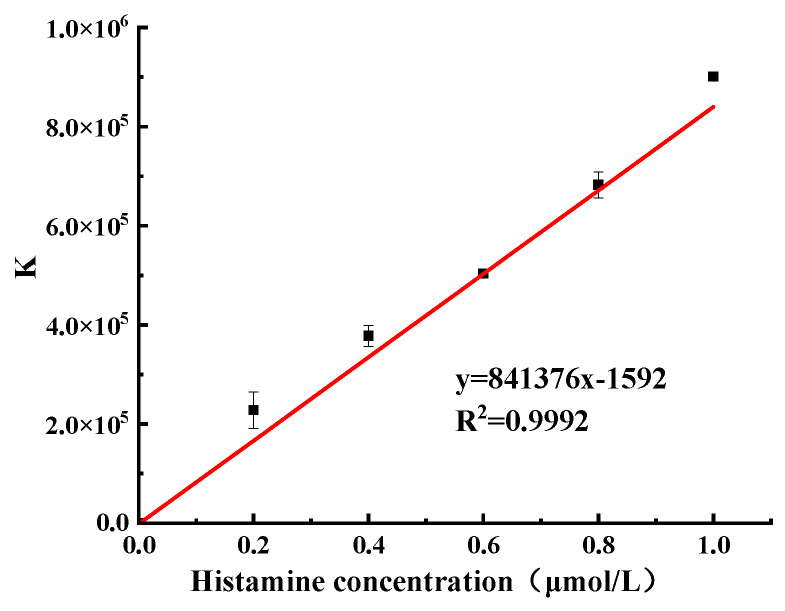
Fluorescence colorimetric dual-mode integrated value (*K*) and histamine concentration.

**Figure 4 molecules-27-08711-f004:**
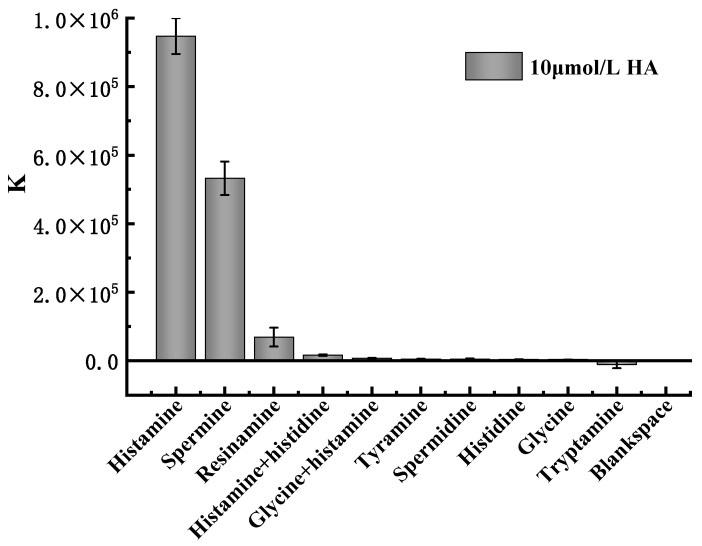
Specificity evaluation of the detection system based on the K signal value.

**Figure 5 molecules-27-08711-f005:**
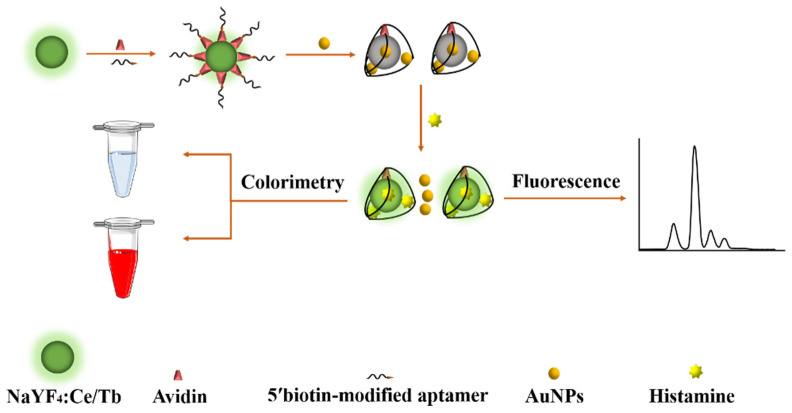
Schematic diagram of histamine detection based on the aptamer/AuNPs fluorometric and colorimetric dual-mode detection.

**Table 1 molecules-27-08711-t001:** Precision analysis of the fluorometric (*ΔF*) determination results.

Histamine Concentration (μmol)	Measurement Results (*n* = 3) (10^5^)			$\bar{x}$ (10^5^)	S (10^5^)	RSD/%
0	0	0	0	0	0	0
0.2	2.0	2.1	2.2	2.2	7.5	3.4
0.4	3.3	3.4	3.6	3.5	9.7	2.7
0.6	5.1	5.0	5.3	5.2	2.0	3.8
0.8	6.5	6.7	6.6	6.7	9.1	1.4
1.0	8.3	8.1	8.4	8.2	2.1	2.6

**Table 2 molecules-27-08711-t002:** Precision analysis of the absorbance value (A) determination results.

Histamine Concentration (μmol)	Measurement Results (*n* = 3)			$\bar{x}$	S (10^−3^)	RSD/%
0	0.50	0.49	0.50	0.49	9.1	1.8
0.2	0.58	0.56	0.59	0.58	16.4	2.8
0.4	0.64	0.65	0.66	0.65	0.4	0.1
0.6	0.73	0.73	0.75	0.74	16.8	2.2
0.8	0.81	0.85	0.87	0.86	14.1	1.6
1.0	0.87	0.87	0.89	0.88	7.8	0.8

**Table 3 molecules-27-08711-t003:** Comparison of different histamine detection methods.

Methods	Linear Range	Detection Limit	Reference
Indirect competitive ELISA method	20~1350 μmol/L	4500 nmol/L	[28]
Non-enzymatic unmodified graphene electrode based method	40~900 μmol/L	5580 nmol/L	[29]
Molecularly imprinted polymer-based fluorescence assay	1.8~44.98 μmol/L	1800 nmol/L	[30]
Plasmonic nanoparticle-based surface plasmon resonance electrochemiluminescence of polyacid-bipyridyl ruthenium	1~1000 μmol/L	100 nmol/L	[31]
Colorimetric	0.2~1.0 μmol/L	69.37 nmol/L	This work
Fluorescence	0.2~1.0 μmol/L	9.21 nmol/L	This work
Dual mode (K-value)	0.2~1.0 μmol/L	4.57 nmol/L	This work

**Table 4 molecules-27-08711-t004:** Recovery rate of the fish samples.

Samples	Amount Added (μmol)	Detection Amount	Recovery (%)
** *Trichiurus haumela* **	0	-	0
	0.2	0.166 ± 0.003	83.41 ± 0.016
	0.4	0.371 ± 0.021	92.87 ± 0.051
	0.6	0.564 ± 0.044	94.02 ± 0.072
	0.8	0.816 ± 0.070	101.94 ± 0.087
	1.0	0.949 ± 0.069	94.98 ± 0.068
** *Thamnaconus septentrionalis* **	0	0.075 ± 0.0138	-
	0.2	0.164 ± 0.010	82.24 ± 0.047
	0.4	0.389 ± 0.005	97.41 ± 0.012
	0.6	0.636 ± 0.010	105.92 ± 0.017
	0.8	0.759 ± 0.30	94.97 ± 0.036
	1.0	0.919 ± 0.04	91.88 ± 0.040

## Data Availability

Not applicable.

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
