# Peer review of "A Dual-Mode Method Based on Aptamer Recognition and Time-Resolved Fluorescence Resonance Energy Transfer for Histamine Detection in Fish"

_molecules, 2022, doi:10.3390/molecules27248711_

Round 1
Reviewer 1 Report
The authors claim that their approach is highly sensitive to fluorescence analysis and the colorimetric detection advantage of simple operation and rapid reaction (line 90-92). While reading the materials and methods, I realized that their methods need expensive chemicals and reagents, and sophisticated equipment is involved in getting the results along with hours of preparation for analysis. I do not think that currently proposed analysis system is economical and time-saving compared to the gold standard methods e.g., HPLC, ELISA etc.
Authors mention that “Meanwhile, the combined detection of the two modes showed good linearity with histamine concentration, the linear detection range of the dual-mode synthesis was 0.2-1.0 μmol/L, with a detection limit of 4.57 nmol/L”. (line 26-28). They have also written that acceptable histamine content in different countries e.g. US FDA 50mg/kg, EFSA 100-200 mg/kg, China 40mg/100g. Per my understanding, the detection limit of the currently proposed system (0.013 mg/dL) is far less than the international standards. So, how is this system practical?
Title:
The title of the manuscript can be improved, and it’s subject (Histamine Detection In Fish) should be placed properly
Abstract:
1. The authors should mention that the sensing system is for detecting histamine in fish meat or fish-based food products.
Introduction:
1. The authors provided different standards of the permissible histamine content in fish meat (line 47-52). It is suggested that the unit (mg/kg vs. mg/g) should be consistent to convince the reader.
2. Aptamers are widely used in food safety detection (lines 67-68). Authors are required to provide references for this claim.
3. Please provide the complete terminology and the abbreviation e.g., SELEX technology (line 71).
Results and discussion
1. I think that section 2.1 can be moved to the materials and methods section.
2. How many times did the authors perform the experiments to get the results of Figure 2 (b)
3. Authors should adequately label all the wells in Figure 2 (c & d)
4. Please explain the unit on the y-axis of the graphs presented in Figure 3.
5. What is the unit of Histamine concentration in Table 1 ?
6. Authors should be consistent in the units provided in Table 3 to avoid misconceptions in the reader’s mind.
7. I strongly suggest that authors must mention the limitations of their study.
Materials and Methods:
1. Correct the chemical formula of Trisodium Citrate (Line 257
2. The spellings of “phosphatebufferedsaline” need correction. (line 279).
3. What is the subject of the sentence? (line 299-301)
4. Please explain the meaning of “final data R”. (line 323)
5. How Trichiurus lepturus and Thamnaconus modestus were sourced?
Conclusion
1. The conclusion seems to be a replication of the abstract.
2. Authors should provide the implications of their findings.
Author Response
- The authors claim that their approach is highly sensitive to fluorescence analysis and the colorimetric detection advantage of simple operation and rapid reaction (line 90-92). While reading the materials and methods, I realized that their methods need expensive chemicals and reagents, and sophisticated equipment is involved in getting the results along with hours of preparation for analysis. I do not think that currently proposed analysis system is economical and time-saving compared to the gold standard methods e.g., HPLC, ELISA etc.
Reply: Thanks a lot for your suggestion. In response to your comment that our materials are more expensive and time consuming, we did a comparison of methods. It was found that HPLC took longer and more complicated than our method. Because it must require sample derivatization, which not only is tedious but also cause more effects to affect the accuracy of quantification. ELISA experiments require antibodies that are difficult and expensive to prepare, and experiments results are tend to be false-positive and unstable. Our method provides a highly selective and sensitive environment for the detection of histamine in fish meat by taking the combined application of fluorescence and colorimetric modes, reducing interference and detection time compared with HPLC and ELISA.
- Authors mention that “Meanwhile, the combined detection of the two modes showed good linearity with histamine concentration, the linear detection range of the dual-mode synthesis was 0.2-1.0 μmol/L, with a detection limit of 4.57 nmol/L”. (line 26-28). They have also written that acceptable histamine content in different countries e.g. US FDA 50mg/kg, EFSA 100-200 mg/kg, China 40 mg/100g. Per my understanding, the detection limit of the currently proposed system (0.013 mg/dL) is far less than the international standards. So, how is this system practical?
Reply: Thanks very much for the suggestion. The low detection limit indicates the high sensitivity of the method. Our method has a wide linear range (0.2~1.0 μmol/L), which covers all international and domestic limit standards and can be applied to the actual detection; while the low detection limit enables our system to achieve high sensitivity detection. Both of these two aspects prove the powerful analytical performance of the assay.
Title:
- The title of the manuscript can be improved, and it’s subject (Histamine Detection In Fish) should be placed properly
Reply: Thank you very much for your constructive comments. We have improved the title: A dual-mode method based on aptamer recognition and time-resolved fluorescence resonance energy transfer for histamine detection in fish.
Abstract:
- The authors should mention that the sensing system is for detecting histamine in fish meat or fish-based food products.
Reply: Thank you very much for your constructive advice. We have added some information in the revision. (Pages 1, Lines 28-30)
Introduction:
- The authors provided different standards of the permissible histamine content in fish meat (line 47-52). It is suggested that the unit (mg/kg vs. mg/g) should be consistent to convince the reader.
Reply: Thank you very much for your constructive advice. We have made these corrections in the revision. (Page2, Lines 51-52).
- Aptamers are widely used in food safety detection (lines 67-68). Authors are required to provide references for this claim.
Reply: Thanks a lot for your suggestion. We added this in the revision. (Page2, Lines 71-76).
- Please provide the complete terminology and the abbreviation e.g., SELEX technology (line 71).
Reply: Thank you very much for your constructive comments. We added this in the revision. (Page2, Lines 78-79).
Results and discussion
- I think that section 2.1 can be moved to the materials and methods section.
Reply: Thank you very much for your constructive comments. In the revision, we have moved the principles into the materials and methods section. (Page9, Lines 270).
- How many times did the authors perform the experiments to get the results of Figure 2 (b).
Reply: Thank you very much for your question. Our experiments were repeated with three times, and the more representative results were shown for presentation. (Page3, Lines 129, Figure 1(b)).
- Authors should adequately label all the wells in Figure 2 (c & d).
Reply: Thanks a lot for your suggestion. We have made additions in the revision. (Page3, Lines 128, Figure 1(c & d)).
- Please explain the unit on the y-axis of the graphs presented in Figure 3.
Reply: Thank you for your question. The Y-axis is the recovery value of fluorescence, which is the differential of fluorescence intensity obtained from the fluorescent material with histamine added.
- What is the unit of Histamine concentration in Table 1 ?
Reply: Thank you very much for your constructive comments. We have made revisions in the table. (Page5 &6, Lines 178-179, Table 1 & Table 2).
- Authors should be consistent in the units provided in Table 3 to avoid misconceptions in the reader’s mind.
Reply: Thanks a lot for your suggestion. We added it into the revision. (Page7, Lines 209, Table 3.)
- I strongly suggest that authors must mention the limitations of their study.
Reply: Thank you very much for your constructive comments. We have added these in the revision. (Page11, Lines 376-379).
Materials and Methods:
- Correct the chemical formula of Trisodium Citrate (Line 257.)
Reply: Thank you very much for pointing out the error. We have made a correction in the revision. (Page8, Lines 252).
- The spellings of “phosphatebufferedsaline” need correction. (line 279).
Reply: Thank you very much for pointing out the error. We have made a correction in the revision. (Page10, Lines 325).
- What is the subject of the sentence? (line 299-301).
Reply: Thank you very much for your constructive comments. We have made a correction in the revision. (Page10, Lines 320-321).
- Please explain the meaning of “final data R”. (line 323).
Reply: Thank you very much for your constructive comments. In order to distinguish the correlation coefficient R, we have corrected it as K in the text and explained it in Section 3.4.5; and in Section 2.3 we presented the equation for K based on the experimental results. (Page5, Lines 174-175; Page11, Lines 351-354).
- How Trichiurus lepturus and Thamnaconus modestus were sourced?
Reply: Thank you very much for your constructive comments. We have investigated a literature review and corrected the scientific names of the two fish used in the experiments purchased from supermarkets. (Page11, Lines 356-357).
Conclusion
- The conclusion seems to be a replication of the abstract.
Reply: Thank you very much for your constructive comments. We have modified the conclusions. (Page11, Lines 369-384).
- Authors should provide the implications of their findings.
Reply: Thank you very much for your constructive comments. We add the significance of the study to the conclusion. (Page11, Lines 376-384).
Reviewer 2 Report
This manuscript details the development of a new fluorescent/colorimetric dual mode sensor for histamine based on fluorescence recovery due to the replacement of gold nanoparticles near an aptamer surface. The authors compare the measurements results with this method with several other standard techniques. They also investigate interferents and finally demonstrate their technique on actual fish samples.
Overall, this is a well written manuscript that is clear and straightforward. This author recommends publication with some minor revisions that are detailed below:
1. Figure 1 (Page 3): Please have Fluorescence on one line
2. Figure 3a (Page 4): Can additional points be included between 30 and 50 nmol/L of the aptamer? Or is there an explanation/reason that can be provided why there is a rapid rising peak at 40 nmol/L followed by the same decrease back to a near constant value?
3. Figure 3b (Page 4): Why do longer incubation times (> 35 minutes) show a decline in delta F? Could points at longer times (> 50 minutes) be added to show if it has stabilized (as mentioned on lines 169-172)?
4. Figure 4 (Page 5): This data looks as if it can be fit with a higher-order curve (not a first order equation). As shown, the calibration line seems to under predict the R values. Could this also be why the values calculated in Table 4 (assuming they are using the equation generated in Figure 4) are also under predicted?
5. Page 5, Equation 1: Is there a reference for this equation? If so, it probably should be added to line 177.
6. Table 1 (Page 5): Are the measurement results columns R or delta F? It looks like they are R. Also, is there really 9 significant digits for these measurements? And could they be cast in scientific notation to make it easier to read?
7. Table 3 (Pages 7-8): It is recommended to change This Job to This Work. Also, can the entries for Plasmonic nanoparticle-based surface plasmon resonance and Electrochemiluminescence of polyacid-bipyridyl ruthenium be cast as mol/L to facilitate direct comparison with the other entries?
8. Table 4 (Pages 8-9): Are the Detection Amounts presented in this table calculated using the equation generated in Figure 4? If so, this could be the reason that the Detection Amounts are under reporting.
9. The Supplemental Information has good information that will be useful for the reader, but some sort of reference to the information should be provided throughout the text, or at least a summary provided in the paper. Without this, the reader will not know what to find in the Supplemental Information.
Author Response
- Figure 1 (Page 3): Please have Fluorescence on one line
Reply: Thank you very much for pointing out this mistake. We have made a correction in the revision. (Page 10, Lines 298, Figure 5)
- Figure 3a (Page 4): Can additional points be included between 30 and 50 nmol/L of the aptamer? Or is there an explanation/reason that can be provided why there is a rapid rising peak at 40 nmol/L followed by the same decrease back to a near constant value?
Reply: Thank you very much for your constructive comments. In the optimal experiment, a fixed amount of AuNPs and histamine were added to figure out the optimal aptamer concentration range. Since the amount of AuNPs added was constant, the amount of inducer binding to AuNPs and the resulting fluorescence quenching did not increase with increasing inducer concentration, while the amount of histamine added was constant and the fluorescence recovery caused by histamine. This is why the higher the concentration of inducer between 20-40 nmol/L, the higher the value of fluorescence recovery appears for the quenched fluorescent material caused by histamine. After reaching the maximum value of histamine binding to the aptamer (40 nmol/L), the concentration of the aptamer increases while the concentration of histamine remains unchanged, so the fluorescence recovery value decreases. The concentration of aptamer at 60 nmol/L is so high that the initial quenching of fluorescence by the aptamer by AuNPs is difficult to recovery compared to few AuNPs. Consequently, the fluorescence recovery value decreases due to the binding of histamine to the aptamer.
- Figure 3b (Page 4): Why do longer incubation times (> 35 minutes) show a decline in delta F? Could points at longer times (> 50 minutes) be added to show if it has stabilized (as mentioned on lines 169-172)?
Reply: Thank you very much for your question. The target histamine causes fluorescence recovery by competing with AuNPs to bind the aptamer, forming a relatively stable relative equilibrium state in which histamine binds to the aptamer and AuNPs aggregate. During the reaction time after the optimal incubation time, the relative equilibrium state gradually changes with increasing reaction time, and the degree of binding of histamine, AuNPs and the aptamer changes, resulting in a decrease in the fluorescence recovery value. An excellent reaction process can be achieved in 40 min incubation time, but too long incubation time will lead to excessive waste of time.
- Figure 4 (Page 5): This data looks as if it can be fit with a higher-order curve (not a first order equation). As shown, the calibration line seems to under predict the R values. Could this also be why the values calculated in Table 4 (assuming they are using the equation generated in Figure 4) are also under predicted?
Reply: Thank you very much for your constructive advice. In the figure, our data have an R2 of 0.9992, which already has a good linear correlation. We also performed a higher-order fit, but the curve effect does not match the reference and practical need for linearity. It is suggested to take the original equation for the calculation.
- Page 5, Equation 1: Is there a reference for this equation? If so, it probably should be added to line 177.
Reply: Thanks a lot for your suggestion. This equation was obtained by referring to relevant literature and has been marked in the revision. (Pages 5, Lines 174-176).
- Table 1 (Page 5): Are the measurement results columns R or delta F? It looks like they are R. Also, is there really 9 significant digits for these measurements? And could they be cast in scientific notation to make it easier to read?
Reply: Thanks a lot for your suggestion. The measurement results are the recovery values of fluorescence, and those errors have modified in the modified version. (Page 5, Lines 180, Table 1)
- Table 3 (Pages 7-8): It is recommended to change This Job to This Work. Also, can the entries for Plasmonic nanoparticle-based surface plasmon resonance and Electrochemiluminescence of polyacid-bipyridyl ruthenium be cast as mol/L to facilitate direct comparison with the other entries?
Reply: Thank you very much for your constructive advice. We have made changes in Table 3 in the revision. (Page 7, Lines 211, Table 3)
- Table 4 (Pages 8-9): Are the Detection Amounts presented in this table calculated using the equation generated in Figure 4? If so, this could be the reason that the Detection Amounts are under reporting.
Reply: Thanks a lot for your suggestion. The data in Table 4 are obtained from calculations using this equation, and the recovery is within ± 20% of the requirements, so we still need to choose the original equation for the calculation.
- The Supplemental Information has good information that will be useful for the reader, but some sort of reference to the information should be provided throughout the text, or at least a summary provided in the paper. Without this, the reader will not know what to find in the Supplemental Information
Reply: Thank you very much for this constructive comment. We have added the relative information in the revision. (Page 5, Lines 175-177; Page 7 Lines 211-212; Page 10, Lines 310-311; Page 10, Lines 313-314)
Reviewer 3 Report
This manuscript describes a fluorescence/colorimetric dual-mode method for the determination of histamine in fish based on the recovery of fluorescence signals resulting from binding between aptamers and histamine. The proposed method is simple but lacking in innovation, and some supplements are needed to enrich the article:
1. The rapid development of electrochemical instruments’ portability makes them suitable for rapid field detection, no less than fluorescence and colorimetry. Please revise the relevant parts of the introduction after consideration.
2. In part “Detection principle”, the principle of the colorimetric method is not explained resulting in the dual mode not being intuitively displayed.
3. In Table 1, 2, and 4, Please specify the unit.
4. In Table 3, the unit should be unified for comparison.
5. In Table 4, The blank Thamnaconus modestus sample detection data lacks error.
6. Table 2, The S`xr calculated according to the RSD seems redundant.
7. The integrated value (R) and correlation coefficient R2 is confused.
8. Figure 2(a) should add the fluorescence signal with AuNPs for comparison.
9. Please figure out In Figure 2(b), is it characteristic surface plasmon resonance (SPR) peak?
10. In Analysis of specificity, the method may had not good selectivity due to fluorescence intensities spermine were 56.22% of the histamine.
Author Response
- The rapid development of electrochemical instruments’ portability makes them suitable for rapid field detection, no less than fluorescence and colorimetry. Please revise the relevant parts of the introduction after consideration.
Reply: Thank you very much for this constructive comment. We have modified the introductory section. (Page 2, Lines 59-66)
- In part “Detection principle”, the principle of the colorimetric method is not explained resulting in the dual mode not being intuitively displayed.
Reply: Thank you very much for the advice. We have added information in the detection principle section. (Page 9-10, Lines 285-297)
- In Table 1, 2, and 4, Please specify the unit.
Reply: Thank you for this comment. We have made additions in the revision. (Page 5 & 8, Lines 180-181 & 245, Table 1 & 2 &4)
- In Table 3, the unit should be unified for comparison.
Reply: Thank you very much for this constructive comment. We have made some additions in the revision. (Page 7, Line 211 , Table 3)
- In Table 4, The blank Thamnaconus modestus sample detection data lacks error.
Reply: Thank you very much for this constructive comment. We have made additions in the revision. (Page 8, Lines 245 , Table 4)
- Table 2, The S`xr calculated according to the RSD seems redundant.
Reply: Thank you for this comments. We have removed the redundant calculated values. (Page 5-6, Lines 181 , Table 2)
- The integrated value (R) and correlation coefficient R2 is confused..
Reply: Thank you very much for the advice. In order to distinguish R2, we changed the R that is derived from the ratio of fluorescence recovery and absorbance into K, and corrected the correlation coefficient in the text. (Page 5, Lines 170-171 , Figure 3)
- Figure 2(a) should add the fluorescence signal with AuNPs for comparison.
Reply: Thank you very much for the suggestion.. We conducted the experiment and added the results in Figure 1. (Page 3, Lines 129 , Figure 1(a))
- Please figure out In Figure 2(b), is it characteristic surface plasmon resonance (SPR) peak?.
Reply: Thank you very much for the suggestion. We have confirmed the scheme and reviewed the relevant literature. The characteristic surface plasmon resonance peaks of AuNPs are shown in Figure 1(b).
- In Analysis of specificity, the method may had not good selectivity due to fluorescence intensities spermine were 56.22% of the histamine.
Reply: Thank you very much for your question. Spermine, as a biogenic amine analogue, is structurally similar to histamine, and thus it is able to cross-react with the aptamer of histamine for structural recognition, causing a change in signal that can be the basis for acting as a multi-target. Our experiments reached quantitative analysis through structural recognition, and the high fluorescence intensity of spermine indicates that the method is more specific and can be used for the specific detection of histamine.
Round 2
Reviewer 3 Report
No more comments.